# The Use of Polydialkylsiloxanes/Triflic Acid as Derivatization Agents in the Analysis of Sulfur-Containing Aromatics by “Soft”-Ionization Mass Spectrometry

**DOI:** 10.3390/molecules27238600

**Published:** 2022-12-06

**Authors:** Zhanna Starkova, Valentina Ilyushenkova, Nikolay Polovkov, Daria Voskressenskaya, Ilya Pikovskoi, Mikhail Tebenikhin, Ella Vtorushina, Anastasiia Kanateva, Roman Borisov, Vladimir Zaikin

**Affiliations:** 1A.V. Topchiev Institute of Petrochemical Synthesis, Russian Academy of Sciences, 29 Leninskiy Prosp., 119991 Moscow, Russia; 2N. D. Zelinsky Institute of Organic Chemistry, Russian Academy of Sciences, 47 Leninsky Prosp., 119991 Moscow, Russia; 3Institute of Medicine, Peoples’ Friendship University of Russia (RUDN University), 6 Miklukho-Maklaya Str., 117198 Moscow, Russia; 4Core Facility Center ‘Arktika’, Northern (Arctic) Federal University, 17 nab.Severnoy Dviny, 163002 Arkhangelsk, Russia; 5V.I. Shpilman Research and Analytical Center for the Rational Use of the Subsoil, 2 Studencheskaya Str., 628007 Khanty-Mansiysk, Russia; 6Department of Plastics, D. Mendeleev University of Chemical Technology of Russia, 9 Miusskaya Pl., 125047 Moscow, Russia; 7Organic Chemistry Department, Peoples’ Friendship University of Russia (RUDN University), 6 Miklukho-Maklaya Str., 117198 Moscow, Russia

**Keywords:** dibenzothiophenes, derivatization, polydialkylsiloxanes, sulfonium salts, mass spectrometry, high resolution, MALDI, ESI, CID

## Abstract

Polycyclic aromatic sulfur-containing compounds are widely distributed in oil, especially in its low-volatile and heavy fractions (resins, asphaltenes), and this dictates the need for their determination when reliable methods for sulfur removing, cleaning and processing oil are developed. In these cases, “soft” ionization mass spectrometry methods, based on electrospray ionization (ESI) and matrix-assisted laser desorption/ionization (MALDI), are particularly effective. However, aromatic sulfur-containing compounds have low polarity and cannot be readily ionized by these methods. To overcome the problem, their preliminary conversion into sulfonium salts by the action of alkyl iodides and a silver-containing agent is widely used. In the process of developing more economical derivatization methods, we found a rather unexpected possibility of implementing S-alkylation of organic sulfides with commercial polydialkylsiloxanes (alkyl = CH_3_ or C_2_H_5_) in the presence of triflic acid (CF_3_SO_3_H) as a superacid co-alkylating agent. For homologous dibenzothiophenes as a typical model representative of petroleum sulfur-containing aromatic compounds, ESI and MALDI mass spectra exhibited the signals of corresponding S-alkylsulfonium salts with a high signal-to-noise ratio. A rational mechanism for the described chemical transformation is proposed, including the indispensable activation by triflic acid and the cleavage of the Si-C bond. Specific collision-induced dissociation of corresponding S-alkylated sulfonium cations is considered. The applicability of the derivatization approach to the analysis of petroleum products by high-resolution mass spectrometry is demonstrated.

## 1. Introduction

Petroleum is a very complex mixture of various types of organic compounds, including hydrocarbons, nitrogen and sulfur heterocyclic compounds. In many respects, such sulfur-containing substances are harmful in the practical use of petroleum products, and knowledge of their qualitative and quantitative composition is extremely important in developing effective methods for removing sulfur from crude oil in order to ensure the high performance of such products [1,2]. There is a number of approaches allowing determining the total content of sulfur in petroleum and petroleum products, including X-ray methods [3] and atomic spectroscopy [4]. However, these methods deal with the whole samples and cannot be applied for molecular characterization of sulfur-containing compounds in petroleum because of the complexity of the samples. An obvious solution to the problem is the application of chromatographic techniques allowing the detection of individual molecules. For example, gas chromatography (GC) and, particularly, comprehensive two-dimensional GC are very efficient in the separation of volatile and semi-volatile compounds of petroleum [5]. Their combinations with selective detectors, such as, for example, sulfur chemiluminescence detector [6] and flame photometric detector [7], or with a more universal mass selective detector [8], allow the identification of a wide range of sulfur compounds. Unfortunately, GC techniques are limited by the volatility of analytes and are not suitable for high molecular weight and polar components of crude oil. In 2000, J. Fenn and D. Zhahn demonstrated that these substances could be detected by electrospray ionization (ESI) [9]. The registered mass spectrum of crude oil contained hundreds of ion signals corresponding to thousands of homologous, isobaric and isomeric substances. The interpretation of these data required the application of ultrahigh-resolution FT-ICR mass spectrometry that was laid on the basis of a new era of petroleum analysis—petroleomics [10].

In general, petroleomics is based on the application of soft ionization mass spectrometry methods, mainly ESI, though the role of other atmospheric pressure and desorption/ionization techniques such as (matrix-assisted) laser desorption/ionization ((MA)LDI) is rapidly growing [11]. The ionization mechanisms of these methods are different. In the case of ESI, for example, ions performed in solution transferred into the gas phase [12], whereas, in the case of MALDI, ions can be formed both during co-crystallization with matrix compounds or in the course of gas-phase ion-molecule reactions in plume [13]. Nevertheless, almost all the processes of ion formation under these conditions involve protonation, cationization or deprotonation of analytes. Therefore, they do not usually provide satisfactory ionization efficiency in the direct detection and identification of weakly polar and nonpolar organic compounds because the ionization of analytes by both of these methods requires performing either cationization (most often protonation) or deprotonation processes. To overcome this shortcoming, preliminary chemical modification (derivatization) is widely used to ensure the introduction of a covalently bonded group, providing a high ionization efficiency, into molecules of analytes [14].

One of the groups of compounds that require indispensable preliminary derivatization prior to mass spectrometric investigation are molecules containing non-polar or weakly polar sulfide or sulfhydryl groups.

To increase the detectability of such compounds, preliminary modifications of targeted sulfur atoms appeared to be promising approaches, among which fixed-charge [15] and charge-generation derivatization are particularly efficient [16]. The simplest version of such reactions includes the interaction of dialkyl sulfides, saturated cyclic sulfides and mercaptans with alkyl halides or trialkyloxonium tetrafluoroborates that proceeds under mild conditions giving rise to sulfonium salts with a quantitative yield [17]. Recently [18], we have shown that the S-alkylation of saturated sulfides can proceed efficiently under the action of alkyl formates in the presence of triflic acid (TfA, CF_3_SO_3_H), which has superacid properties. In all these cases, sulfonium salts are the end products which easily desorb ready-made sulfonium cations under ESI and MALDI conditions and provide a sharp increase in the ionization efficiency of the compounds to be analyzed.

The situation becomes more complicated when it is necessary to derivatize aromatic sulfides for further mass spectrometric investigation. Unlike aliphatic sulfides and saturated thiacyclanes, sulfur-containing aromatic compounds practically do not readily undergo the S-alkylation reactions described above. Meanwhile, polycyclic aromatic sulfur-containing compounds are widely distributed in oil, especially in its high molecular weight fractions (resins, asphaltenes), and this strongly dictates the need for their qualitative and quantitative determination when efficient methods for cleaning and processing oil are developed [1,2]. To solve this derivatization problem, S-methylation or ethylation of aromatic sulfides by corresponding iodides in the presence of a silver salt as a co-alkylating agent to form corresponding sulfonium salts was suggested. In the first work describing this principle, S-methylation by means of methyl iodide in the presence of AgBF_4_ was suggested [19], and this procedure was used in a large number of works dealing with high-resolution (Fourier transform ion cyclotron resonance) mass spectrometric analysis of petroleum sulfides (see, for example, [20]). Note that relatively recently, another co-alkylating agent, namely AgSbF_6_, was proposed for the same purposes [21].

During the development of new methods for chemical pre-modification in the study of variously functionalized compounds by “soft” ionization mass spectrometry, we found that sulfur-containing organic compounds can be effectively converted to sulfonium salts in the presence of polydialkylsiloxanes and triflic acid (TfA, CF_3_SO_3_H)) as a co-alkylating agent. It is especially important that this reaction turned out to be applicable not only to dialkyl sulfides and saturated thiacyclanes but also to typical aromatic sulfides of the dibenzothiophene series. The present work is focused on investigating the possibilities of using such preliminary derivatization for MALDI and ESI mass spectrometric determination of aromatic sulfides. Available polydimethylsiloxane (PDMSO) and polydiethylsiloxane (PDESO) of the general structure R_3_Si-O-[SiR_2_-O]_n_-SiR_3_ (R = CH_3_ or C_2_H_5_) were tested as S-alkylating agents for these purposes. Dibenzothiophene and its homologues were used as model compounds. In addition, the fragmentation features of corresponding sulfonium cations, which can be useful in structural and quantitative determinations, were established by tandem mass spectrometry. The effectiveness of the derivatization approach was also demonstrated in the example of the analysis of real oils by ultrahigh-resolution mass spectrometry.

## 2. Results

### 2.1. Optimization of Derivatization Procedure

In the present work, model dibenzothiophene (DBT) and its methyl- and ethyl-substituted homologues (3–4), which are typical structural representatives of sulfur-containing aromatics in heavy oil fractions, were taken (Figure 1). Silicone fluids based on PDMSO and PDESO were used as potential derivatization agents that can be activated by superacid TfA.

It should be immediately noted that the proposed derivatization reaction is carried out in the absence of an additional solvent, the role of which is performed by the silicone fluid itself. For this reason, polydialkylsiloxane as a derivatizing agent is always present in the reaction mixture in large excess, and hence the yield of products does not depend on the concentration of sulfides.

Using individual benzothiophenes and varying the reaction temperature in the range of 30 to 100 °C, we found that the maximum yield of products is achieved at a temperature of 80 °C if the reaction is carried out for 10–40 min. A further increase in temperature and reaction time does not affect the yield of reaction products, but at the temperature of about 100 °C unidentifiable decomposition products of polysiloxanes appear due to the presence of superacidic TfA in the mixture. Therefore, we chose the temperature of 80 °C as the most optimal in further derivatization experiments.

In all cases, corresponding S-alkyl dibenzothiophenium salts (triflates or acetates) were the products of studied sulfur-containing aromatics. The structure of derivatives was confirmed by NMR spectroscopy and high-resolution mass spectrometry data. The reaction yields were evaluated by analysis of polysiloxanes medium after derivatization procedures by gas chromatography-mass spectrometry (GC-MS). Only traces of initial compounds were detected, confirming that the reaction proceeded almost quantitatively and with a high degree of selectivity, regardless of the structure of the starting sulfide.

### 2.2. Possible Mechanism of the Described Derivatization Reaction

In general, the observed derivatization properties of polydialkylsiloxanes seem rather unexpected since the Si-alkyl bond energy is not sufficiently low to imply the possibility of its breaking in some exchange process. However, examples of silicon-carbon bond scission in the presence of strong acids have already been described [22,23]. In our case, the possibility of S-alkylation of sulfides is almost certainly due to the presence of such a superacid as TfA, which is most likely an important initiator of the reaction and should be considered a co-alkylation agent. The most probable mechanism should involve heterolytic cleavage of the Si-C bond with the transfer of a proton from the SO_3_H group to the silicon atom and the release of corresponding alkyl cation, which can easily interact with sulfur compounds yielding corresponding salts [24]. We suppose that a similar process occurs in forming a cyclic sulfonium cation, the counter-ion for which in the corresponding sulfonium salt is the anionic residue of TfA (Figure 2). The expected difference in the stability of liberated methyl or ethyl carbo-cations does not practically affect the yield of desired S-alkyl dibenzothiophenium salts under the described optimal reaction conditions.

### 2.3. Features of MALDI and ESI Mass Spectra of S-Alkyldibenzothiophenium Cations Extracted from Salts

Since the ionization processes of organic compounds under MALDI and ESI conditions are largely similar, one could expect the registration of fairly identical mass spectra. In fact, this is also the case for S-alkyl dibenzothiophenium salts, as is clearly confirmed by comparing the MALDI and ESI mass spectra recorded for the same salt (Figure 1). Taking into account this fact, all further discussion will be carried out mainly on the example of mass spectra recorded during the ESI.

In all cases, regardless of the structure of starting model dibenzothiophene and the nature of the derivatization agent (PDMSO/TfA or PDESO/TfA), the analyzed salts exhibited well-reproducible MALDI and ESI spectra with a sufficiently high signal-to-noise ratio for the targeted ions of interest. The *m/z* values for the heaviest ions showed that they have 15 and 29 Da higher mass than the molecular weight of the initial aromatic sulfides in the case of PDMSO and PDESO, respectively. These single mass increments indicate that the detected ions are due to dibenzothiophenium cations bearing S-methyl or S-ethyl substituent.

Figure 2 and Table 1 present the ESI mass spectra recorded for the derivatization products prepared from an artificial mixture of all four dibenzothiophenes using PDMSO/TfA (a) and PFESO/TfA (b) as alkylation agents. To the initial mixture, sulfur-containing thianthrene and dibenzyl sulfide were also added for comparison. Judging by the peak ratio of the target ions, the yields of C-alkylated sulfonic salts were practically independent of the nature of the alkylating agent. This shows that both PDMSO and PDESO reagents could be equally used for the analysis of not only petroleum aromatic sulfides but also sulfur-containing compounds of other structures and origins.

Earlier [5], for the case of S-alkylsulfonium cations originating from S-alkylated saturated thiacyclanes during MALDI and ESI, we showed that their collision-induced dissociation (CID) gives rise to the mass spectra containing a significant amount of information about the structure of sulfides. In the case of the S-alkylated dibenzothiophenium cations, there is no need to extract specific information about their structure from CID spectra. At the same time, these data could be useful in the quantitative determination of the corresponding sulfides in complex mixtures using the reaction monitoring approach. Therefore, we will consider the features of CID mass spectra only for the case of sulfonium cations generated under ESI conditions.

Figure 3 and Figure 4, Table 2 present the examples of CID mass spectra of some S-methyl(ethyl) dibenzothiophenium cations revealing most typical fragmentation pathways. Dissociation of S-methyl dibenzothiophenium cations is rather simple and begins with the detachment of the CH_3_ radical, giving rise to cation radicals of corresponding dibenzothiophenes (Figure 3a–c). The latter ions most likely have the structure of primary ionization ions generated from initial substituted DBT under electron ionization. In fact, corresponding ions only slightly decompose in the case of S-methylated cations of DBT (Figure 3a) and eliminate H or CH_3_ radicals (at the expense of ethyl substituent) in the case of 4MDBT or 4,6DMDBT (see ions with 197 and 225 Da in Figure 3b,c respectively).

The situation changes in the case of S-ethyl cations. Here, ethyl substituent at the S atom can be eliminated in the form of both radical and ethylene neutral. As a result, the pairs of peaks at *m*/*z* 184/185 and 198/199 are observed in CID spectra recorded for S-ethyl cations corresponding to DBT and 4MDBT. As expected, the ion at 199 Da in the latter case loses methyl substituent as radical (compare Figure 4a,b). CID of S-ethyl-46DEDBT cation starts from the elimination of ethyl radical attached to sulfur (ion 240 Da) followed by the loss of C-ethyl groups or their fragments (Figure 4c).

### 2.4. Example of Application of Preliminary Derivatization by Poly(dialkyl)siloxanes to Analysis of Petroleum Sulfur-Containing Aromatics by High-Resolution ESI Mass Spectrometry

The proposed approach was tested for the detection of sulfur compounds in crude oil. A sample from the Romaskinskoye oilfield was deasphaltenized and derivatized using alkylation by poly(dialkyl)siloxanes and the traditionally used method of alkylation by CH_3_I/AgBF_4_. The product of derivatization was analyzed using ultrahigh-resolution mass spectrometry with ESI ionization. The comparison of the results clearly shows that the proposed approach allows the detection higher number of compounds in a wider range of DBEs and numbers of carbon atoms. To visualize such a comparison, isoabundance contours plots of DBE vs. carbon number for the S_1_ heteroatom class were built (Figure 5) using DropMS software (http://www.dropms.online/ (accessed on 21 November 2022)) [25].

## 3. Materials and Methods

### 3.1. Chemicals

Model dibenzothiophene (DBT), its 4-methyl (4MDBT), 4,6-dimethyl (46DMDBT) and 4,6-diethyl (46DEDBT) homologs, as well as thianthrene and dibenzyl sulfide, were from Sigma-Aldrich (St. Louis, MO, USA). Silicone fluids based on polydimethylsiloxane PMS-100 (PDMSO) (from Reakhim, Moscow, Russia; kinematic viscosity 100 cSt) and polydiethylsiloxane (PDESO) (from Gelest Inc., Morrisville, PA, USA, kinematic viscosity 80 cSt) were used as solvents and derivatizing reagents. Triflic acid (TfA) with a purity of 99% (from Fluorochem, Glossop, UK) was used as an alkylation activator. 1,8,9-Antracenetriol (AT) as a MALDI matrix was from (Aldrich Chemical Co., Hoeilaart, Belgium). Glacial acetic acid with a purity of 99.7% (from Panreac, Castellar del Vallès, Spain), chemically pure n-hexane, toluene and tetrahydrofuran (from Himmed, Russia) were used. Experiments with crude oil were performed using soar medium petroleum (total sulfur content 2.5% wt., density 0.882 g/cm^3^ at 20 °C) produced at Romashkinskoye oilfield (Berezovskaya, well 32,941, depth 1707–1726 m).

### 3.2. Derivatization of Dibenzothiophenes and Their Analogues by Silicon Fluids

A solution of 30 μg of individual dibenzothiophenes (or a mixture of homologues and analogues) in 700 μL of PDMSO or PDESO was placed in a 2 mL glass vial and heated to 80 °C with stirring on a vortex mixer. About 20 μL of TfA were added to the resulting homogeneous mixture, covered with a lid and stirred on a vortex mixer for another 40 min at 80 °C. A sign of the reaction was the turbidity of the solution and the formation of drops of an isolated colored liquid at the vial bottom. The reaction mixture was cooled to room temperature, and 300 µL of glacial acetic acid was added, thoroughly mixed and centrifuged at 6000 rpm. Using a pipette, the bottom layer containing triflates of S-methyl or S-ethylsulfonium salts was transferred to a 2 mL plastic vial and washed three times with hexane to remove the remainder of the silicone fluid.

### 3.3. Derivatization of Dibenzothiophenes by Silicon Fluids and Characterization of the Derivatives

A solution of 50 mg of dibenzothiophene in 10 mL of PDMSO or PDESO was placed in a 20 mL flask vial and heated to 80 °C, and stirred for 20 min. About 300 μL of TfA was added to the resulting homogeneous mixture and stirred intensively for 40 min at 80 °C. A sign of the reaction was the turbidity of the solution and the formation of drops of an isolated colored liquid at the vial bottom. The reaction mixture was cooled to room temperature, and l ml of glacial acetic acid-d3 was added. Using a pipette, the bottom layer containing triflates of S-methyl or S-ethylsulfonium salts was transferred to a 2 mL plastic vial and washed three times with hexane to remove the remainder of the silicone fluid. The residue was characterized by 1H NMR and HRMS.

#### 3.3.1. S-Methyldibenzothiophenium Triflate

1H NMR (acetonitrile-d3) δ 3.8 (s, 3H), 7.7 (t, 2H), 7.9 (t, 2H), 8.2 (m, 4H). HRMS ESI (+) (*m/z*): C13H11S, 199.0579 (found), 199.0576 (calc).

#### 3.3.2. S-Ethyldibenzothiophenium Triflate

1H NMR (acetonitrile-d3) δ 1.6 (t, 3H), 4.40 (q, 2H) 3.8 (s, 3H), 7.8 (t, 2H), 7.9 (t, 2H), 8.2 (d, 2H), 8.3 (d, 2H). HRMS ESI (+) (*m/z*): C14H13S, 213.0734 (found), 213.0732 (calc).

### 3.4. Sample Preparation for Analysis by MALDI Mass Spectrometry

The reaction product was mixed with a solution of the matrix compound AT in tetrahydrofuran (concentration 30 mg/mL) in a ratio of 1:10, and the mixture was applied to a steel target without further purification. After drying, the target was placed in the inlet system of a mass spectrometer to record the MALDI mass spectra.

### 3.5. Sample Preparation for Analysis by ESI Mass Spectrometry

To record ESI mass spectra, the reaction product was diluted 50 times with acetonitrile or methanol and injected into the ionization region using a syringe pump.

### 3.6. Sample Preparation for Analysis by GC/MS

100 μL of the upper layer of the reaction was transferred to a vial and diluted with 2 mL of hexane. A 1 μL solution was injected into the GC/MS system.

### 3.7. Instrumentation

MALDI mass spectra (positively charged ion detection mode) were obtained on a Bruker autoflex speed mass spectrometer (Bruker Daltonics Inc., Leipzig, Germany) equipped with a solid-state UV laser (λ = 355 nm) and a reflectron in the positively charged ions detection mode at the lowest possible laser energy (usually 20% from maximum). To record MALDI mass spectra, an MTP 384 ground steel target (Bruker Daltonics Inc., Bremen, Germany) was used.

ESI mass spectra (positively charged ion detection mode), as well as second-order mass spectra obtained using collision-induced dissociation (CID), were recorded on an Agilent 6470 mass spectrometer (needle voltage 4000 V, capillary voltage 2500 V, nebulizer gas pressure 20.0 psi, spray gas flow 11.0 L/min, spray gas temperature 400 °C, drying gas flow 15.0 L/min, drying gas temperature 300 °C. Nitrogen was used as a collision gas (collision energy 30 eV).

Investigation of oil products prepared after derivatization by high-resolution mass spectrometry was carried out on a Q Exactive Plus mass spectrometer (Thermo Scientific, Waltham, MA, USA) with a quadrupole mass filter and an orbital ion trap mass analyzer with resolving power 120,000 (FWHM, at *m/z* 200). The instrument was equipped with an Ion Max ion source in ESI configuration. Mass scale calibration was carried out daily using a Pierce standard mixture (Thermo Scientific, Waltham, MA, USA) according to the manufacturer’s recommendations to ensure long-term mass accuracy < 3 ppm. In order to eliminate the solvent signals and minimize ion source contamination, the flow injection was used. Each sample injection was carried out in at least three replicates with further averaging of the peak intensities. In all experiments, the following ion source parameters optimized in preliminary tests were used: desolvation capillary temperature 250 °C, S-lens RF voltage e 55 arb. units, curtain gas (nitrogen) flow 2 arb. units, sheath and auxiliary gas (nitrogen) pressure e 25 and 5 psi, respectively. The full spectrum scanning mode with registering signals of positive ions was used in *m/z* ranges 100–1000. The results of at least 10 scans were averaged with further subtracting the background signals of the solvent flow before and after the sample zone. The peak picking procedure was performed using the relative intensity threshold value of 0.1%corresponding to the dynamic range of an Orbitrap mass analyzer (5000:1). Mass spectrometer control, data collection and primary processing were performed using Xcalibur software (Thermo Scientific, Waltham, MA, USA).

GC/MS analysis was performed using Thermo Focus DSQ II instrument (capillary column Varian VF-5ms, length—30 m, I.D.—0.25 mm, df—0.25 μm, carrier gas—helium, injector temperature—270 °C, the initial chromatograph oven temperature—40 °C, heating rate 5 °C/min to 300 °C, followed by isotherm for 10 min; MS parameters: electron ionization with 70eV ionization energy, source temperature—230 °C, selected ion monitoring *m/z* 184, 198, 212, 225, 240, MS detection was stopped after 50 min because of elution of siloxane liquids components).

The NMR spectra were recorded on a Bruker AVANCE III HD spectrometer at 400 MHz (1H NMR) in an acetonitrile-*d3* solution. Chemical shifts δ are reported in parts per million (ppm) relative to the reference (residual acetonitrile-d3 signal).

## 4. Conclusions

In the present work, the possibility of derivatization of aromatic cyclic sulfides by readily available low-viscosity silicone liquids based on polydialkylsiloxanes in the presence of triflic acid as a co-alkylating agent is presented. The reaction proceeds under comparatively mild conditions and could be considered as an alternative to previously developed derivatization approaches to the analysis of weakly polar sulfur-containing compounds by soft ionization mass spectrometry. The essential point in this methodology is that the silicone liquid is not only a derivatization reagent but also a solvent, allowing the reaction to be carried out in a fairly wide temperature range. In addition, these reagents are cheap and readily available.

## Data Availability

Not applicable.

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
