# Peer review of "The Use of Polydialkylsiloxanes/Triflic Acid as Derivatization Agents in the Analysis of Sulfur-Containing Aromatics by “Soft”-Ionization Mass Spectrometry"

_molecules, 2022, doi:10.3390/molecules27238600_

Round 1

Reviewer 1 Report

The manuscript entites The Use of Polydialkylsiloxanes/Triflic Acid as Derivatization
Agents in the Analysis of Sulfur-Containing Aromatics by
“Soft”-Ionization Mass Spectrometry describes the developing of an eco
nomical derivatization method for organic sulfides with commercial polydialkylsiloxans resin
in the presence of triflic acid. The S-alkyl sulophnium products present an enhanced detection by ESI-MS/MALDI. The applicability of the derivatization approach was demonstrated to the analysis of petroleum products by high-resolution mass spectrometry. The interesting paper is well written, and the derivatization reaction is novel.

There is some concerns that should be addressed:

1.      The manuscript lack quantitative data e.g., reaction yields, limit of detection, and validation. How did quantitation made?

2.      What are the expected concentrations of polycyclic aromatic sulfur in petroleum products? And does the method feet the range?

3.      Please provide evidence to support your claim in the results section “optimization of derivatization procedure”

4.      Reaction mechanism. Is there any literature information to support your suggested mechanism?

5.      Scheme 2 in the caption “suggested mechanism for S-alkylation of DBT” in the figure dimethyl sulfide is drawn. Please correct.

6.      Fig. 1 +3 captions. what was the collision energy and concentration of the reaction?

7.      Page 2, line 78. S-methylation(ethylation),  please correct

Author Response

Dear Reviewer,

Thank for your extensive work with our manuscript and valuable comments. Here are our answers for them:

  1. The manuscript lack quantitative data e.g., reaction yields, limit of detection, and validation. How did quantitation made?

The proposed approach, as the most analytical approaches to petroleomics, not quantitative in general. Usually, it is supposed that ionization efficiencies of the derivatives are close enough to compare their intensities (Fig.4). The reaction yields were evaluated by analysis of polysiloxane-containing medium after derivatization procedures by gas chromatography mass/spectrometry. Only traces of initial compounds were detected confirming that the reaction proceeded almost quantitatively.

  1. What are the expected concentrations of polycyclic aromatic sulfur in petroleum products? And does the method feet the range?

In fact, the complexity of crude oil does not allow the assumption to be made about the expected concentrations of polycyclic aromatic sulfur. We can use a number of methods to determine total sulfur in petroleum.  Among them, GC/MS can be used to quantify low molecular weight aromatic compounds. However, polycyclic compounds can be detected only by ‘soft’ ionization methods, which, unfortunately, have a lack of universality and allow detecting only ionizable compounds. As a result, each MS method is applied to discover only a part of the picture.  Therefore, in our case we can just compare the number of detected molecules by the proposed and earlier developed approaches.

  1. Please provide evidence to support your claim in the results section “optimization of derivatization procedure”

As it has been mentioned, we used GC/MS for analysis of polysiloxanes medium after derivatization procedures. The presence of underivatized model compound (DBT) was a signal that the reaction conditions are not optimal and must be improved. We suppose that TIC chromatograms acquired for various reaction temperatures will not be useful for the readers, but we can add them to SI, if the reviewer considers, that it is necessary

  1. Reaction mechanism. Is there any literature information to support your suggested mechanism?

The proposed mechanism based on the described Si-C bond cleavage under action or in presence of strong protic acids [1-3].  For example, in [4] the authors states that ‘Cleavage of the C-Si bond by electrophilic or nucleophilic reagents is well known, and both protic acids and Lewis acidic metal halides can be used as electrophiles’. Such scission Si-C bond lead to formation of the corresponding carbocations, which readily reacts with sulfur compounds [5]. The proposed mechanism confirmed by the formation of ions corresponding to S-alkylated sulfur compounds. Any other alkylation process will not allow detecting such ions. We registered NMR spectrum for purified products of methylation and ethylation of dibenzothiophene and get HRMS data to confirm their elemental composition. The corresponding discussions, reverences and data have been added to the manuscript.

  1. Eaborn, C.; Lickiss, P. D.; Ramadan, N. A. Cleavage of Silicon–Carbon Bonds in Tris(Trimethylsilyl)Methylsilicon Compounds by Trifluoroacetic Acid. Rearrangements and Anchimeric Assistance. J. Chem. Soc., Perkin Trans. 2, 1984, 267–270. https://doi.org/10.1039/p29840000267.
  2. Itami, K.; Terakawa, K.; Yoshida, J.; Kajimoto, O. The Carbon–Silicon Bond Cleavage of Organosilicon Compounds in Supercritical Water. Bulletin of the Chemical Society of Japan, 2004, 77, 2071–2080. https://doi.org/10.1246/bcsj.77.2071.
  3. Lai, Y.-C.; Dvornić, P. R.; Lenz, R. W. Exactly Alternating Silarylene-Siloxane Polymers. 4. Step-Growth Polymerization Reactions with Dichlorosilane Monomers. Journal of Polymer Science: Polymer Chemistry Edition, 1982, 20, 2277–2288. https://doi.org/10.1002/pol.1982.170200830.
  4. Valk, J.-M.; Boersma, J.; van Koten, G. Selective Intramolecular Cleavage of the Carbon-Silicon Bond by Palladium Salts. Journal of Organometallic Chemistry, 1994, 483, 213–216. https://doi.org/10.1016/0022-328x(94)87166-3.
  5. Aggarwal, V. K.; Thompson, A.; Jones, R. V. H. Synthesis of Sulfonium Salts by Sulfide Alkylation; an Alternative Approach. Tetrahedron Letters, 1994, 35, 8659–8660. https://doi.org/10.1016/s0040-4039(00)78464-4.

6. Scheme 2 in the caption “suggested mechanism for S-alkylation of DBT” in the figure dimethyl sulfide is drawn. Please correct.

In fact, we tried to draw common structure for all cyclic sulfur compounds and used dotted arc to show cycles. We agree that this scheme can be misunderstood and corrected it.

  1. Fig. 1 +3 captions. what was the collision energy and concentration of the reaction?

Collision energy was 30 eV (this information was presented in the Experimental section), but we agreed that it will be more comfortable to readers to  have this values in Discussion section and have added this information to the caption of the Figures. The concentration of the analytes in course of the model reactions was 30 µg per 700 µl of siloxane medium.

  1. Page 2, line 78. S-methylation(ethylation), please correct

Done

 Best regards,

The authors

Reviewer 2 Report

Review on "The Use of Polydialkylsiloxanes/Triflic Acid as Derivatization Agents in the Analysis of Sulfur-Containing Aromatics by “Soft”-Ionization Mass Spectrometry" by Starkova et al. submitted to "Molecules".

The presented paper describes a new method of derivatisation of polycyclic aromatic compounds containing sulphur for the subsequent identificatia mass spectrometry. While the topic of this paper might have a limited importance and interest for a few specislists, the current draft is inacceptable for publication, and the text should be extensively reworked.

1) The Introduction section does not reflect the state-of-the-art. Moreover, the extensive inappropriate self-citation is detected! Actually, the Introduction is composed in a such way that it only reviews the former papers by the authors, but does not introduce the reader to the problem. The section should be completely rewritten, and the self-citations should be removed.

There are already several methods developed for both the qualitative and quantitative analysis of the heavy fractions of crude oils, including sulphur-containing compounds. These methods include X-ray absorption and X-ray photoelectron spectroscopy, gas and liquid chromatography, and atomic spectrometry. The brief literature survey reveals dozens of papers in the field. These methods and their possibilities for quantitative and qualitative analysis of crude oils should be reviewed in the Introduction.

The references to the description of the ESI and MALDI techniques in mass-spectrometric analysis should be added, so the reader could acquire the necessary information about these ionisation techniques and their strengths and weaknesses.

2) This journal is interdisciplinary and is devoted to the wide audience that should not be experts in the field of mass-spectrometry. Therefore, the presentation of the results should be optimised for the wide audience. More specifically, each presented mass spectrum should be fully described, and the tables, explaining, which ionised fragment corresponds to each mass-to-charge ratio peak should be provided for each spectrum.

3) The mass-spectrometry alone is not sufficient for the unambiguous identification of the molecular fragments. Actually, the authors did not prove the proposed alkylation mechanism. Was the proposed alkylation reaction of aromatic sulphuric compounds studies earlier? Is it common for contemporaty organic chemistry? If yes, it should be pointed out and properly referenced. If the proposed reaction is novel, then it should be studied with the model compound, the product should be purified, separated and properly characterised by at least, the elemental analysis, and the presence of the proposed functional groups.

4) The authors state that "the yield of alkylation reaction was almost quantitative". How was it determined? Which by-products might be formed? How were the samples purified, separated and analysed?

5) Once again, the reader should not be an expert in mass-spectrometry, and therefore, the meaning of the "heat map" (Figure 5) should be explained, and it should be mentiioned, how this figure was produced.

6) What do the "diamond" signs in Figures 3 and 4 mean?

7) Some peaks in Figures 2, 3 and 4 are surrounded by several minor peaks. Does it mean that the alkylation reaction, or the subsequent ionisation has multiple pathways, and the several by-product cations are formed?

8) The method, presented by the authors, is suitable for qualitative analysis only, there are no quantitative data produced. However, even the qualitative methods of analysis should be validated, and their analytical performance parameters, including false positive and negative, sensitivity and specificity rate, cut-off, unreliability region, ruggedness and cross-reactivity should be tested (see 10.1016/S0165-9936(04)00201-8 for example).

9) The word "possess" means primarily "enter into and control, as of emotions or ideas" rather than "own" or "have a property". Its usage in the text is incorrect. Please, rephrase some sentences in the abstract and in the text.

Best regards,

The reviewer.

Author Response

Dear Reviewer,

Thank for your extensive work with our manuscript and valuable comments. Here are our answers for them:

1) The Introduction section does not reflect the state-of-the-art. Moreover, the extensive inappropriate self-citation is detected! Actually, the Introduction is composed in a such way that it only reviews the former papers by the authors, but does not introduce the reader to the problem. The section should be completely rewritten, and the self-citations should be removed.

There are already several methods developed for both the qualitative and quantitative analysis of the heavy fractions of crude oils, including sulphur-containing compounds. These methods include X-ray absorption and X-ray photoelectron spectroscopy, gas and liquid chromatography, and atomic spectrometry. The brief literature survey reveals dozens of papers in the field. These methods and their possibilities for quantitative and qualitative analysis of crude oils should be reviewed in the Introduction.

The references to the description of the ESI and MALDI techniques in mass-spectrometric analysis should be added, so the reader could acquire the necessary information about these ionisation techniques and their strengths and weaknesses.

The Introduction section was prepared keeping in mind that the manuscript would be submitted to Analytical Chemistry section and special issue on Derivatization. So we considered reasonable to limit this section by derivatization of petroleum sulfur compounds issues. However, we absolutely agree with reviewer that our version of introduction does not reflect the state-of-the-art and is not suitable for interdisciplinary journal. The section has been modified to include information concerning other possible methods. The abnormal self-citation can be justified by our extensive work in the field, but we have removed some of our works and extended representation of other publications.

2) This journal is interdisciplinary and is devoted to the wide audience that should not be experts in the field of mass-spectrometry. Therefore, the presentation of the results should be optimised for the wide audience. More specifically, each presented mass spectrum should be fully described, and the tables, explaining, which ionised fragment corresponds to each mass-to-charge ratio peak should be provided for each spectrum.

We suppose that the reviewer’s advice concerns Figures 2, 3 and 4, because mass spectra on Figure 1 contain only one signal. We have added tables clarifying the data and corrected descriptions.

3) The mass-spectrometry alone is not sufficient for the unambiguous identification of the molecular fragments. Actually, the authors did not prove the proposed alkylation mechanism. Was the proposed alkylation reaction of aromatic sulphuric compounds studies earlier? Is it common for contemporaty organic chemistry? If yes, it should be pointed out and properly referenced. If the proposed reaction is novel, then it should be studied with the model compound, the product should be purified, separated and properly characterised by at least, the elemental analysis, and the presence of the proposed functional groups.

The proposed mechanism based on the described Si-C bond cleavage under action or in presence of strong protic acids [1-3].  For example, in [4] the authors states that ‘Cleavage of the C-Si bond by electrophilic or nucleophilic reagents is well known, and both protic acids and Lewis acidic metal halides can be used as electrophiles’. Such scission Si-C bond lead to formation of the corresponding carbocations, which readily reacts with sulfur compounds [5]. The proposed mechanism confirmed by formation of ions corresponding to S-alkylated sulfur compounds. Any other alkylation process will not allow detecting of such ions. However, we agree that such reasoning should be confirmed by other methods. Therefore, we registered NMR spectrum for purified products of methylation and ethylation of dibenzothiophene and get HRMS data to confirm their elemental composition. The corresponding discussions, reverences and data have been added to the manuscript.

  1. Eaborn, C.; Lickiss, P. D.; Ramadan, N. A. Cleavage of Silicon–Carbon Bonds in Tris(Trimethylsilyl)Methylsilicon Compounds by Trifluoroacetic Acid. Rearrangements and Anchimeric Assistance. J. Chem. Soc., Perkin Trans. 2, 1984, 267–270. https://doi.org/10.1039/p29840000267.
  2. Itami, K.; Terakawa, K.; Yoshida, J.; Kajimoto, O. The Carbon–Silicon Bond Cleavage of Organosilicon Compounds in Supercritical Water. Bulletin of the Chemical Society of Japan, 2004, 77, 2071–2080. https://doi.org/10.1246/bcsj.77.2071.
  3. Lai, Y.-C.; Dvornić, P. R.; Lenz, R. W. Exactly Alternating Silarylene-Siloxane Polymers. 4. Step-Growth Polymerization Reactions with Dichlorosilane Monomers. Journal of Polymer Science: Polymer Chemistry Edition, 1982, 20, 2277–2288. https://doi.org/10.1002/pol.1982.170200830.
  4. Valk, J.-M.; Boersma, J.; van Koten, G. Selective Intramolecular Cleavage of the Carbon-Silicon Bond by Palladium Salts. Journal of Organometallic Chemistry, 1994, 483, 213–216. https://doi.org/10.1016/0022-328x(94)87166-3.
  5. Aggarwal, V. K.; Thompson, A.; Jones, R. V. H. Synthesis of Sulfonium Salts by Sulfide Alkylation; an Alternative Approach. Tetrahedron Letters, 1994, 35, 8659–8660. https://doi.org/10.1016/s0040-4039(00)78464-4.

4) The authors state that “the yield of alkylation reaction was almost quantitative”. How was it determined? Which by-products might be formed? How were the samples purified, separated and analysed?

The yield of the reaction was evaluated using model derivatization of dibenzothiophene (DBT). The reaction medium after the separation of the polar layer, containing formed salts, was analyzed by gas chromatography/mass spectrometry. Only traces of DBT were detected and no non-polar byproducts was discovered. The analysis of polar layer by ESI and MALDI mass spectrometry confirmed, that only target derivative formed.  To perform NMR studies the polar layer was rinsed 10 times by hexane and solved in acetone-d6. Details of the experiments have been added to the manuscript.

5) Once again, the reader should not be an expert in mass-spectrometry, and therefore, the meaning of the "heat map" (Figure 5) should be explained, and it should be mentiioned, how this figure was produced.

Heat maps are widely used in petroleomics and other omics to demonstrate dependence between the intensity of the detected signals corresponding to selected class of compounds and other parameters (DBE and the number of carbon atoms in our case). Necessary discussions have been added to the manuscript.

6) What do the “diamond” signs in Figures 3 and 4 mean?

Figures 3 and 4 represent mass spectra of products of collision-induced dissociation of the derivatives. ‘Diamond’ signs point precusor ions. Necessary information have been added to the manuscript.

7) Some peaks in Figures 2, 3 and 4 are surrounded by several minor peaks. Does it mean that the alkylation reaction, or the subsequent ionisation has multiple pathways, and the several by-product cations are formed?

We would like to underline, that Figures 3 and 4 correspond to tandem mass spectra and contain signals corresponding to fragmentation of molecules in CID conditions. These peaks doesn’t prove or disprove formation of by-products. Figure 2 represents ESI mass spectra and minor signals correspond to interaction of the formed derivatives with solvents.

8) The method, presented by the authors, is suitable for qualitative analysis only, there are no quantitative data produced. However, even the qualitative methods of analysis should be validated, and their analytical performance parameters, including false positive and negative, sensitivity and specificity rate, cut-off, unreliability region, ruggedness and cross-reactivity should be tested (see 10.1016/S0165-9936(04)00201-8 for example).

Yes, in general the reviewer is absolutely right. However, in case of petroleomics and some other omics approaches some of such validation procedures are close to be impossible because of the complexity of the matrix. So specificity of the method is based on application of ultra-high resolution mass spectrometry allowing determining elemental compositions of the peaks, whereas sensitivity is determined by the number of the detected compounds.    To support this claims we would like refer to the most cited work in the field of the derivatization of sulfur compounds for their detection in crude oil [Müller, H.; Andersson, J.T.; Schrader, W. Characterization of high-molecular-weight sulfur-containing aromatics in vacuum residues using Fourier transform ion cyclotron resonance mass spectrometry. Anal. Chem. 2005, 77, 2536–2543. https://doi.org/10.1021/ac0483522]. As the review can make sure no specific validation test were performed in these cases.

9) The word "possess" means primarily "enter into and control, as of emotions or ideas" rather than "own" or "have a property". Its usage in the text is incorrect. Please, rephrase some sentences in the abstract and in the text.

The sentences have been rephrased.

Best regards,

The authors

Round 2

Reviewer 2 Report

Dear authors,

Thank you for your revision. Good job!

The reviewer suggests only a couple of additional minor corrections:

1) Please, provide the details of 1H NMR and GC/MS experiments, including the instrumentation and the experimental details (stationary and mobile phases, flow rate etc.).

2) Please, specify the geographic coordinates of the oilfield from where the raw samples were taken, and add the basic description of the crude oil collected from there according to common petroleum classifications by density (light, medium, or heavy) and total sulphur content (sweet or sour).

Best regards,

The reviewer.

Author Response

Dear Reviewer,

Thank you for your supportive comments and valuable suggestions. Here are our answers for them:

1) Please, provide the details of 1H NMR and GC/MS experiments, including the instrumentation and the experimental details (stationary and mobile phases, flow rate etc.).

The necessary description has been added to Materials and Methods section (3.7 Instrumentation)

2) Please, specify the geographic coordinates of the oilfield from where the raw samples were taken, and add the basic description of the crude oil collected from there according to common petroleum classifications by density (light, medium, or heavy) and total sulphur content (sweet or sour).

The necessary description has been added to Materials and Methods section (3.1. Chemicals)

Best regards,

The authors